# Untangling the Complexities of Processing and Analysis for Untargeted LC-MS Data Using Open-Source Tools

**DOI:** 10.3390/metabo13040463

**Published:** 2023-03-23

**Authors:** Elizabeth J. Parker, Kathryn C. Billane, Nichola Austen, Anne Cotton, Rachel M. George, David Hopkins, Janice A. Lake, James K. Pitman, James N. Prout, Heather J. Walker, Alex Williams, Duncan D. Cameron

**Affiliations:** 1School of Biosciences, University of Sheffield, Sheffield S10 2TN, UK; 2Department of Biology, University of Oxford, Oxford OX1 3RB, UK; 3biOMICS Mass Spectrometry Centre, University of Sheffield, Sheffield S10 2TN, UK; 4Department of Earth and Environmental Sciences, University of Manchester, Manchester M13 9PL, UK

**Keywords:** metabolomics, untargeted, mass-spectrometry, open-source, bioinformatics

## Abstract

Untargeted metabolomics is a powerful tool for measuring and understanding complex biological chemistries. However, employment, bioinformatics and downstream analysis of mass spectrometry (MS) data can be daunting for inexperienced users. Numerous open-source and free-to-use data processing and analysis tools exist for various untargeted MS approaches, including liquid chromatography (LC), but choosing the ‘correct’ pipeline isn’t straight-forward. This tutorial, in conjunction with a user-friendly online guide presents a workflow for connecting these tools to process, analyse and annotate various untargeted MS datasets. The workflow is intended to guide exploratory analysis in order to inform decision-making regarding costly and time-consuming downstream targeted MS approaches. We provide practical advice concerning experimental design, organisation of data and downstream analysis, and offer details on sharing and storing valuable MS data for posterity. The workflow is editable and modular, allowing flexibility for updated/changing methodologies and increased clarity and detail as user participation becomes more common. Hence, the authors welcome contributions and improvements to the workflow via the online repository. We believe that this workflow will streamline and condense complex mass-spectrometry approaches into easier, more manageable, analyses thereby generating opportunities for researchers previously discouraged by inaccessible and overly complicated software.

## 1. Introduction

Untargeted metabolomics is an increasingly popular tool for identifying perturbations within a metabolome and revealing phenotypic complexity in systems [1,2,3,4]. It is commonly the first part of a two-step research pipeline, where untargeted studies are used to gather information, identify the metabolome, and generate hypotheses. This is followed by targeted metabolomics which measures specific compounds and requires a priori knowledge of the whole metabolome [1,4,5]. Key to a metabolomics workflow are the data processing and handling steps, which take raw mass spectrometry data and convert them for use in a wide array of multivariate and statistical methods. Currently there is no one standardised pipeline for this step due to variation from sampling methods, instrumentation used, analytical methods employed and the deficit of standardised guidelines [6,7,8,9,10,11]. 

After over a decade of experience with proprietary software, the challenge was to address a number of issues with current common practices and embrace an open-source approach to metabolomics data processing and analysis that can have a future legacy. As well as navigating the plethora of analysis options available, with the advent of remote working, it became apparent that researchers conducting untargeted metabolomics analysis required resources to learn how to process mass spectrometry data remotely. 

The objective of this work was to develop a guide focussed on processing and analysis of mass spectrometry data, collected to address untargeted metabolomics questions, primarily in the fields of environmental metabolomics and the study of complex plant stress responses. However, the tutorial and workflow have been applied in a range of experimental systems including *E. coli*, potato, barley, organic fertilisers, field soil samples, human cervical mucus, and *Chlorella.* The aim is that the guide will help to move towards standardised methodology and comparable research across the field of metabolomics. 

The newly developed workflow presented here is designed to address the question:

Which compounds might be responsible for the difference in metabolomic fingerprint between the classes (groups) of samples?

The workflow converts mass spectrometry data to open formats for experiments in which a wide array of compounds are compared between two or more classes of samples. The steps may not result in a definitive difference or unquestionable compound identification, rather the workflow will direct further research and highlight potential compounds to focus on for targeted analysis. This resource is aimed at non-experts, and early career researchers who may not have extensive coding or analytical knowledge. Users are introduced and guided through pre-processing options and data formatting steps which result in a peak table data frame. This peak table forms the basis of the next steps in the workflow, multivariate analysis and putative metabolite ID to give a list of potential compounds that are differentially expressed between groups of samples which can inform the hypothesis for downstream targeted analyses. Alongside some command-line interface, GUI software has also been utilised in the workflow, which can be simpler to learn and easier to operate for new and non-expert users of metabolomics data analysis software [12]. Notably, all software approaches discussed here are free, as the authors believe it is important that the discussed pipelines are accessible. 

This collaborative and open-source workflow guide for untargeted metabolomics addresses the need for data-handling tutorials [1] with the key aims of widespread use and continuous improvement, ultimately encouraging integration with multi-omic workflows.

## 2. Materials and Methods

### 2.1. Overview and Workflow Diagram

This tutorial guides the user through the untargeted metabolomics workflow that has been developed with some explanation of what each stage achieves. Further details are available in step-by-step guides on the associated website (https://untargeted-metabolomics-workflow.netlify.app/ accessed on (27 January 2023)), which includes links to relevant open-source tools, and our own interoperable code where appropriate. This tutorial covers the steps required to process LC-ESI-MS data, however detailed instructions for processing MALDI-ToF-MS and DI-ESI-MS using similar open-source tools are also available on the associated website.

An index of openly-available datasets is provided at https://untargeted-metabolomics-workflow.netlify.app/00_overview/06_demo-data/ (accessed on 9 March 2023). These example datasets can be used to demonstrate the workflow presented here.

The workflow has been divided into stages. The following number codes are used in the online guide as well as in the R [13] code and workflow diagram (for an abridged version of this diagram see Figure 1).

00. Overviews, workflow diagram & useful information

01. Metabolite extraction

02. Data acquisition (Mass Spectrometry)

03. Converting data to open format

04. Data pre-processing

05. Extracting & formatting peak table & metadata

06. Multivariate analysis (PCA) & further analysis (if applicable)

07. Putative metabolite identification

08. Archiving data & citing resources

Stages 01 and 02 are not covered in great detail in this documentation which focuses primarily on data processing and analysis.

### 2.2. Experimental Design and Quality Control

Difficulties in analysis and/or workflows can arise from complexities in experimental structure. Many terms are used interchangeably in different contexts. Most tools for untargeted metabolomics are set up for one factor analysis with two or three levels e.g.,

Case vs. controlWild-type vs. transgenic lineStrain 1 vs. strain 2 vs. strain 3

However, more complex experimental designs are quite often implemented e.g.,

Two factors with two or more levels in each such as +/− treatment for two strainsTime course for one or two factors such as +/− treatment for two strains over three time points

To begin, the expectations of which groups of metabolite fingerprints may differ from one another must be considered, and to what extent.

What are the biological replicates being analysed and are they independent of each other (or has the same organism/population been sampled multiple times)?Are there technical replicates (i.e., repeated runs of the same sample)?Are Quality Control (QC) samples required? Are analytical standards needed?What groupings are required to answer the research questions outlined?

Quality control (QC) can mean different things to researchers from different fields. There are a few simple quality control options for checking that there has not been subtle (or not so subtle) variation accumulating during the run. Decisions must be made on which one (or more) of these are necessary depending on the type of sample to be analysed and the MS techniques employed:Spike all prepared samples with a compound for which the *m*/*z* (and RT) is known and which is unlikely to be otherwise present in the experimental samples;Prepare a pooled QC sample from an aliquot of each of the samples and include this at regular intervals in the MS run;Include blanks and/or extraction blanks at regular intervals in the MS run;Use lock mass calibration (for Waters instruments).

There are some basic data quality control steps you can take to limit errors during processing and analysis:Check file sizes of .raw files across the MS run;Check file sizes of converted .mzML files—reconvert any that are unexpected;Compare spectra between technical replicates

### 2.3. Metabolite Extraction and Data Acquistion

Details of quenching, metabolite extraction or choice of mass spectrometry platform are not covered here, as they will likely be specific to the organism and/or tissue involved and the questions being addressed. Figure 2 provides a conceptual overview of metabolite extraction and data acquisition from plant tissues. See [14,15] for introductory guidance and [16] for a specific metabolite extraction method appropriate to plant tissues for this workflow.

### 2.4. Preparing Metadata for Analysis

To process and analyse data using our workflow, two .csv files are required (these can be created in excel, R, google sheets etc. depending on preference) as long as the order and headings of the columns follow the pattern detailed below.

For samplelist.csv the following columns are required:“Filename”: this is a list of the filenames of the .mzml files (the part before the .mzml)“Filetext”: this is the name that has been manually added to the metadata of that sample“MSFile” or an equivalent column that contains either “pos” or “neg” within it. Any other columns will be ignored in this file.

For treatments.csv at least two columns are required:
“Filetext”: this must contain all the distinct values of “Filetext” from samplelist.csv“Variable1”: the naming of this column is left to the user. For example, in an MS run comparing a wild-type to a control, this column could be named “treatment” and filled with “WT” and “C” as appropriate“Variable2” etc: further variables. This may include batch identifiers (for example if many samples were run over multiple days), treatments or environmental variables

These are kept in a folder with the .mzml data files. Examples can be found on the website at https://untargeted-metabolomics-workflow.netlify.app/03_conversion-to-open-format/05_samples-treatments/ (accessed on 27 January 2023).

## 3. Results

### 3.1. Converting Data to Open Format Using Proteowizard

Converting proprietary data files (which contain a large amount of data and metadata about the run in separate files) to a more manageable format, such as .mzML (the standard open-data format for mass spectrometry [17]) is essential. We have developed this workflow using .RAW files, which are specific to Waters software and are not compatible with many open-source tools. To convert .RAW to .mzML, Proteowizard software [18] is used. Proteowizard is capable of converting many other proprietary file formats and guidance is available through their extensive documentation at https://proteowizard.sourceforge.io/doc_users.html accessed on (20 February 2023). Proteowizard comprises two applications: SeeMS and MSConvert.

SeeMS is useful for viewing chromatograms and spectra without access to proprietary software like MassLynx. MSConvert performs conversion of the MS data but depending on the type of MS used, different settings/parameters in MSConvert may be required, detailed in the online step-by-step instructions to complete stage 03 (https://untargeted-metabolomics-workflow.netlify.app/03_conversion-to-open-format/03_msconvert-lcms/ accessed on 27 January 2023).

It is critically important to check the size of .mzML files once converted. They should all be similar. SeeMS can be used to check any that seem unusual and reconvert any with an incongruous file size (problems in conversion can arise, for instance from intermittent internet connection when converting files from a remote drive).

### 3.2. Preprocessing Data

Untargeted metabolomics datasets can be several GB in size! To get from compressed .mzML files to a tractable peak table that can be interrogated with multivariate statistics, it is necessary to “tidy” the data.

A peak table is a data-frame consisting of aligned spectra with concentration or intensity values against a set of features—mass to charge ratio (*m*/*z*) or *m*/*z* with retention time (RT). The file size will be dependent on sample number but will be smaller than the .mzML files.

Different downstream tools for multivariate statistics will require the peak table in slightly different formats, so the code included in this guide will help with formatting for some common uses (e.g., MetaboAnalyst one factor and two factor peak tables) as well as helping format treatment information as metadata so that peak tables can be interrogated.

Depending on the MS approach, different stages are involved but they broadly fall into:Baseline correction and/or noise reduction (estimating what part of the detected intensity is the sample and “cleaning” or adjusting the spectra to show only the signal believed to be associated with the sample);Normalisation and/or standardisation (these can mean a range of different things to different people but broadly cover accounting for differences in sample volume or concentration or total intensity of the signal);Grouping and peak picking (wave-form algorithms are used to determine which parts of the spectra constitute separate peaks utilising their *m*/*z* value);Alignment or peak matching (assessing across samples to determine whether peaks with slightly different *m*/*z* values are the same peak so that samples can be compared more reliably).The above criteria are very important when processing data as they can have a big impact on data quality however the parameters may vary with different datasets and different analysis methods. The importance of these factors have been discussed previously by [19].

By the end of this stage, data will be processed into a single table containing all the *m*/*z* and intensity values required for down-stream analysis. This stage relies on the use of open-source software (XCMS online [20] for LC-ESI-MS and MassUp [21] for MALDI-ToF-MS and DI-ESI-MS) to process the data. These provide user interfaces for well-documented R packages (XCMS [22] and MALDIquant [23] respectively) and provide the advantage of coping well with large datasets and, in the case of XCMS online, being run remotely.

For detailed instructions on pre-processing, consult stage 04 of our online guide (https://untargeted-metabolomics-workflow.netlify.app/04_data-preprocessing/ accessed on (27 January 2023)).

R code to extract a peak table from pre-processed data is available in stage 05 of our online guide (https://untargeted-metabolomics-workflow.netlify.app/05_extracting-formatting-peak-table/ accessed on (27 January 2023)).

### 3.3. Multivariate Analysis

There are often two key questions when analysing a new untargeted metabolomics dataset:Are the metabolomic fingerprints distinct classes (treatment groups) different from each other?Which features of the metabolomic fingerprint are causing them to be different from each other?

To answer the first question, data ordination is required to provide a global overview of the variability and patterns within the data. Principal Component Analysis (PCA) is a commonly applied ordination tool that reduces the dimensionality of multivariate data to display complex relationships between samples in 2 or 3 dimensions [15]. As it is unsupervised the model is unaware of the classes to which the samples belong, so patterns are unbiased by a priori knowledge of the experimental design. PERMANOVA can be used to provide statistical corroboration of patterns observed in the PCA by statistically evaluating if significant trends exist at the higher levels of the experimental design within multivariate data i.e., if significant treatment and interaction effects are present. Finally, where clear differences between classes in the PCA are apparent, pairwise comparisons between classes (treatment groups) can be investigated via exploring the loadings or using a pairwise analysis such as t-tests or volcano plots. These will provide the user with features of interest that are most important at defining the statistical output [15]. 

Where patterns are less clear, supervised analysis, such as OPLS-DA (orthogonal projections of latent structures) may be employed to mine for differences between any two classes. The output of supervised analyses will highlight particularly highly abundant features that differ between two randomly assigned classes that may be obscured in global overview if the majority of the metabolome is conserved or unchanging (this can occur in tissues where only small numbers of metabolites respond to a stimulus, but the majority of the metabolome is unaffected). To limit false positives it is important to consider the native separation in the data (i.e., through an unsupervised ordination, like PCA) to provide a robust biological justification for comparing two particular classes. The analyses exemplified here are by no means the only option, and it is highly recommended that tools such as MetaboAnalyst [24] are employed by the researcher to explore all analytical avenues available.

In the online guide, demonstration is given on how to perform these analyses using a free online platform and how to run some alternative code in R. MetaboAnalyst is an online platform on which untargeted metabolomics data can be loaded, normalised, analysed and visualised. However, there is a strong emphasis on detailed statistics that may be more appropriate for targeted analyses, so the user must have a clear understanding of their objectives in choosing amongst the options. 

MetaboAnalyst is interoperable with R and the underlying code can be accessed using the button at the top left of the “Results” page. The advantage of running the code is that the user can integrate it with other analyses (and formatting for figures). Examples of figures produced with this approach can be found in Figure 3. In contrast, the advantage of the MetaboAnalyst GUI is that it guides the user through the process and has some useful sense-checks and vignettes available.

Details can be found via the excellent tutorials and documentation provided by MetaboAnalyst [25].

It is also possible to analyse the same peak tables using SIMCA (Umetrics) or other proprietary softwares. However, it is much harder (and more costly) to use these remotely, and it is harder to document any analysis for sharing with other researchers. Other software worth considering includes MSDial, MetaboKit and MeV [26,27,28].

### 3.4. What Are My Metabolites?

It is very important to consider that this stage of the metabolomic process is not automated and can be incredibly time-consuming and challenging to do, so it is advisable that the preceding analysis has been adequately assessed for its effectiveness before committing time at this stage.

Annotating metabolomic features is challenging—there are some automated annotations included with e.g., XCMS that rely on the CAMERA package [29] amongst others. However, these often struggle with unusual experimental structures and/or large datasets, or “unusual” (i.e., non-human) metabolites. Thus, reducing the number of metabolomic features to those that are causing a significant (in terms of reliability and magnitude) difference between two classes of samples is advisable.

To ascertain the identity of these features, comparing the *m*/*z* (or *m*/*z* at specific RT) values highlighted by multivariate analysis with databases of reference *m*/*z* and with experimental data from the literature (usually available in a publication or in repositories like MetaboLights [30] and Metlin [31]) is key.

Stage 07 of the online guide provides guidance on using a range of databases to help annotate “metabolites of interest” (https://untargeted-metabolomics-workflow.netlify.app/07_putative-metabolite-id/ accessed on (27 January 2023)). These include:METLIN to search by *m*/*z*;KEGG PATHWAY and KEGG COMPOUND [32] to corroborate likelihood of detecting certain compounds in the study organism/sample and to gain insight on biological function;Data repositories such as MetaboLights;Details of how to find other relevant databases (MassBank, PubChem, MetaCyc, Metabolomics Workbench [33,34,35,36]);Reporting Metabolomics Standards Initiative (MSI) identification levels (see also [37]).

### 3.5. Sharing Metabolomics Data

Metabolomics data from even a small study can be very large. It can also be very complex. But there are ways of sharing it with the wider scientific community (and indeed the public) without too much trouble. It is insufficient to only prepare a data availability statement or simply share graphs or peak tables.

Metabolomics data can be analysed in lots of different ways, so it is important to comply with the FAIR principles [38]:FindableAccessibleInteroperableReusable

Institution-based data repositories are an option, but they often require extra levels of support to submit large datasets and there is no guarantee that access to other researchers is feasible.

More useful is a field-specific repository where data will be made available together with other relevant data sets. Furthermore, these repositories provide guidance on appropriate data formatting, allowing it to be compatible with other published data to form part of potential future meta-analyses. Some journals will have specific guidelines on which repository to use [39].

Time should be set aside from the outset of any project for submitting data to a repository. It is not optional!

MetaboLights is a data repository specific to metabolomics studies [30]. Data from NMR, GC-MS, LC-MS, and MALDI amongst others, may be submitted.

The repository is maintained and curated by the European Bioinformatics Institute (EMBL-EBI) meaning that the data it holds is well-formatted and integrated with several other standardised databases and ontologies (ways of describing methods, data and metadata). This “future-proofs” the data stored, making it not only open-access but also more findable and reusable, as well as facilitating integration with other -omics data, if required.

MetaboLights has various stages of submission, validation and then curation by experts to make sure each submission has all the relevant metadata needed to recreate the analysis undertaken. Following curation, there is a review process and finally data can be added to the repository and made available.

Because of the curation process, there can be a significant lag between submission and data being available so early submission is advisable. However, once submitted, there is a reference that can be linked to any publication [30].

Account creation is required, after which, a video tutorial guide on using the submission portal is available. Additional hints and tips on this can be found on the associated website (https://untargeted-metabolomics-workflow.netlify.app/08_data-archiving-citation/02_metabolights/ accessed on (27 January 2023)).

### 3.6. Citation of the Tools Used in the Workflow

Links to cite the following tools involved in the workflow can be found at https://untargeted-metabolomics-workflow.netlify.app/08_data-archiving-citation/03_citing-tools/ accessed on (21 February 2023). These tools are regularly updated so it is important to cite the version used and/or the date accessed:All R packages used;R and RStudio versions;Proteowizard (SeeMS and MSConvert);Metaboanalyst;XCMS online and METLIN;MassUp;MassBank (including access date);ECMDB and any other organism specific metabolite databases used;KEGG (including BRITE, COMPOUND and PATHWAY);PubChem;A data availability statement that links to your archived data (e.g., in MetaboLights).

## 4. Conclusions

At this point the choice in preparing and analysing metabolomics data is at the discretion of the research group. This guide is a useful starting point that leads the reader through an openly available, best-practice, pipeline. Complex data and analytical processes can be overwhelming, but by engaging in discussion forums, sharing ideas, troubleshooting, and having access to a community of like-minded researchers these processes can become more accessible and facilitate exploration of exciting biological questions.

## Figures and Tables

**Figure 1 metabolites-13-00463-f001:**
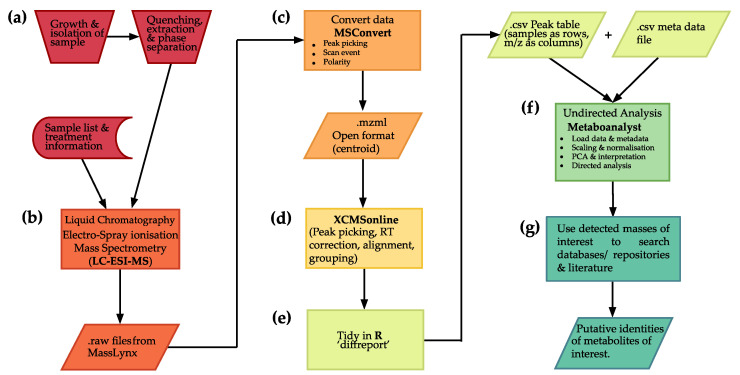
Workflow diagram for processing and analysis of untargeted LC-MS metabolomics data. (**a**) sample selection and preparation. (**b**) Mass spectrometry analysis of samples. (**c**) Conversion of data to open format. (**d**) Data pre-processing and (**e**) production of a feature matrix with experimental information included. (**f**) Statistical analysis for selection of features of interest and (**g**) identification of features of interest by comparison with literature and existing metabolite databases.

**Figure 2 metabolites-13-00463-f002:**
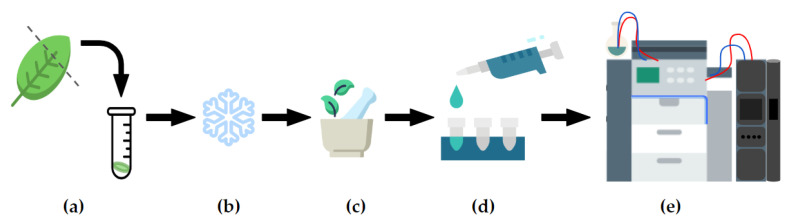
Conceptual diagram of an untargeted metabolomics workflow, from leaf to mass spectrometry analysis. After sample harvest (**a**), metabolic reactions in a sample tissue must be first quenched (**b**); i.e., via liquid nitrogen immersion), cell walls lysed and the sample homogenised (**c**) to permit extraction of compounds within the cells using a range of solvents (**d**). Extracts may then be diluted and submitted to mass spectrometry analysis (**e**); e.g., UPLC-ESI-MS).

**Figure 3 metabolites-13-00463-f003:**
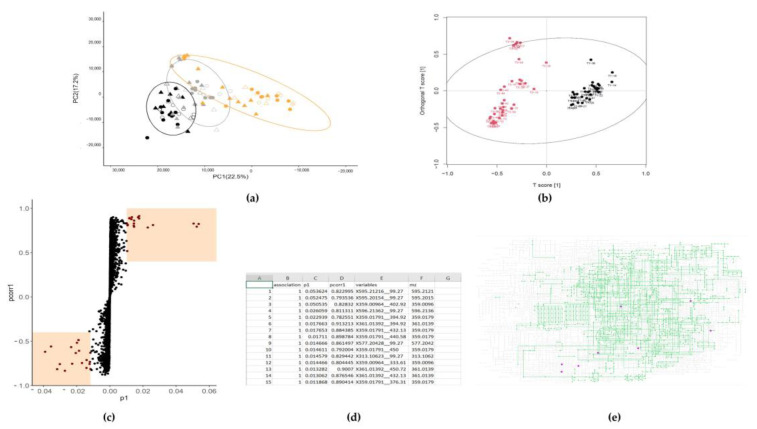
Conceptual diagram of examples of multivariate analysis outputs of untargeted metabolomics analysis, all produced using open-source or freely available software. (**a**) Principal component analysis (PCA) 2-D scores plot produced with *pcaMethods* and *ggplot2* packages in R; (**b**) OPLS-DA scores plot produced using the *muma* package in R; (**c**) scores plot created using *ggplot2* package and data produced by the *muma* package in R; (**d**) example list of features of interest highlighted by an OPLS-DA using *muma* in R; (**e**) example of metabolites highlighted within a KEGG pathways global *Esterichia coli* metabolism map.

## Data Availability

No new data were created or analysed in this study. Research software described in this article is available online https://github.com/LizzyParkerPannell/Untargeted_metabolomics_workflow accessed on (27 January 2023). The associated online guide is available at https://untargeted-metabolomics-workflow.netlify.app/ accessed on (27 January 2023).

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
