# Peer review of "Untangling the Complexities of Processing and Analysis for Untargeted LC-MS Data Using Open-Source Tools"

_metabolites, 2023, doi:10.3390/metabo13040463_

Round 1

Reviewer 1 Report

The article is devoted to a really important topic, the workflow for analysis of untargeted metabolomics data. However, the authors did not disclose the main idea of the article.

Major:

1. The authors list all the necessary stages of data analysis workflow, but do not discuss their importance and possible solution ways, especially for normalization and alignment.

2. Since the authors use different open sources for analysis, it is worth to pay more attention to the description of the capabilities of these sources and comparing them with others.

Minor: 

1. The Section 2.2 and references 12-16 are not relevant and can be omitted. 

Author Response

We thank reviewer 1 for their recognition of our contribution to this important topic, and for their useful feedback.  Taking on board the point that the main idea of the article was not abundantly clear, we have restructured the introduction to give a clearer framing and contextualisation of our work. Addressing the major points raised:

  1. In section 2.6 the necessary stages of data analysis are discussed, however the paper is designed as an introduction to our workflow and many review papers extensively cover the topics of data preprocessing, filtering, normalisation, standardisation and scaling. We provide detailed instructions on these pre-processing steps in the online guide (which we regularly signpost toward in this manuscript), which presents comprehensive discussion of these parameters. However, for the sake of clarity, we have included a useful reference that details these criteria and their importance (line 222-225 of revised manuscript).

  2. As this manuscript is an introduction to our workflow, further detail is available on the online guide. While comparing and contrasting different open-source packages available for metabolomic analysis is outside the scope of this manuscript, we agree with the reviewer’s point that such a comprehensive comparative chart would be useful to help navigate the ‘best’ approach to processing one’s data. We currently link to a number of other helpful guides and resources from the site (e.g. https://untargeted-metabolomics-workflow.netlify.app/00_overview/02_stages-of-workflow/) but intend to add further options of open-source tools to the workflow as time permits. Since the resource is a community effort, we welcome contributions via the github repo (https://github.com/LizzyParkerPannell/Untargeted_metabolomics_workflow). 

In answer to minor point 1: While we feel this section provides valuable background information to the workflow before the data processing, we have re-worded it for clarity and succinctness, as well as trimmed the references.This balance will aid the reader to grasp the entire analytical process and sign-post to useful resources for further learning.

Reviewer 2 Report

Parker et al. present a tutorial for conducting data analysis with LCMS analytics. MS data analysis is challenging and the authors managed to construct a valuable workflow for training and reassurance of the different steps. The main strength of the educational workflow is the associated website and the manuscript can be seen as the teaser to increase the visibility of the online content. The workflow with all explanations is available on GitHub with a permissive CC-By license, which allows community development of the workflow as well as specific lab adaptations via local installations. While the main idea of the workflow is well-developed, it should be improved on a couple of aspects before publication.

1. The first data conversion steps rely on GUI-based software tools. While these are attractive to experimentalists, please also introduce some terminal/programming oriented software from R and Python, e.g. pymzml, omigami, pyMassSpec.

2. The step in the online version '02. Data acquisition' is empty and only links to the previous 'Metabolite Extraction' step and can be removed.

3. There should be example data available and the tutorial should stay close to the sample data.

4. The introduction could be streamlined to the standard format of background, problem, and solution.

5. It would be great to have some Jupyter Notebooks available for the R-based analysis steps.

6. The GitHub page should contain keywords/labels, a short description, and a link to the website. On the main website, the 'Edit this page' link leads nowhere.

7. How specific is the tutorial to the Waters instrumentation and MS in general? The admirable open attitude is diminished somewhat if the tutorial is restricted to a particular proprietary.raw file type. Thermo also generates a .raw file, what are the differences? After all, the steps subsequent to the open format are independent of the source.

Specific Line-associated suggestions:
L105: When mentioning R/RStudio, there should be a citation, or a link on how to get the software.
L162 (caption 2.1.6), 237: Please provide a more objective formulation
L132,146, 255: Please avoid direct speech to address the reader in the manuscript. This is fine on the webpage with a more colloquial setting but sounds odd in the manuscript.
E.g.: The user should come up with clear questions which should be addressed by the analysis and should determine in advance how the data should be classified.
L179: Twice ‘for example’ in a single sentence.
L221 & 223: Please specify ‘huge’ and ‘a little bit’
L235: cleaning away -> cleaning
L267, 273: Please provide a more thorough introduction to PCA and OPLS-DA, including scores plots and classes, perhaps even with a sketch.
L400: Discussion forums and communities are a good point and more concrete internet information would be great.

Minor:
a. In the supplement the figure sequence is mixed.
b. The capitalization in bullet points is incoherent.

Author Response

We thank reviewer 2 for their careful consideration of the manuscript. We are grateful for their acknowledgement of the purpose of the manuscript as a “teaser” to the online guide, and appreciate them taking the time to explore the website. We value the feedback we have received from reviewer 2 and have addressed their comments as follows:

  1. This tutorial is aimed primarily at those users who have limited experience with bioinformatics, and as such, we have added justification for using GUI-based software tools in the introduction (lines 64-67 of the revised manuscript). In a number of places the online workflow directs users to R packages that can achieve the same procedure as the chosen GUI, and by choosing XCMSonline and Metaboanalyst we allow a scaling up of skills, because the R code is accessible alongside the GUI view. We thank reviewer 2 for the suggestions of alternative conversion tools and have updated the website to signpost the user to available script-base packages that can perform data format conversions (https://untargeted-metabolomics-workflow.netlify.app/03_conversion-to-open-format/01_proteowizard/).

  1. On the website, further information about data acquisition has been added, including some brief comparison of analytical methods (https://untargeted-metabolomics-workflow.netlify.app/02_data-acquisition/01_data-acquisition/). Links to improve navigation between sections have also been added (this step in the online version was previously empty).

  2. Example MALDI data (https://doi.org/10.5061/dryad.dbrv15f5c) has been published on DRYAD, an open-access repository and this has been linked to on the website. Our own example LC-MS data is currently being prepared for submission to DRYAD. In the meantime, on the website we have linked to an example dataset that can be accessed through MetaboLights (https://untargeted-metabolomics-workflow.netlify.app/00_overview/06_demo-data/). 

  3. The introduction has been reformatted and restructured to briefly discuss the background, problem, then identifies the workflow as the solution and explains how the workflow is to be implemented (lines 32-73 of the revised manuscript). 

  4. While formatting the workflow as a Jupyter Notebook is beyond the scope of this publication the aim of the version-controlled online workflow is to constantly evolve. We hope to include Jupyter Notebooks as part of this in the future (and would welcome any contributions via github https://github.com/LizzyParkerPannell/Untargeted_metabolomics_workflow).

  5. Keywords, a short description and a link to the website have been added to the github “about” section as advised. The “edit this page” button which led nowhere has been removed (the github repository already has a direct link in the menu shortcuts).

  6. This section has been edited to include “Proteowizard is capable of converting many other proprietary file formats and guidance is available through their extensive documentation at https://proteowizard.sourceforge.io/doc_users.html accessed 20th February 2023.” Earlier emphasis of the use of Waters/ MassLynx specific steps has been removed and mentioned only as examples.  As reviewer 2 highlights, the steps subsequent to the open format are independent of the source of the data.

L105 (R citation): though the location of the first mention of the R language/ environment has moved, we have now included a reference for this.

Throughout the manuscript we have removed direct speech (for example, in the section “Experimental design and quality control” on line 106 and the section “citation of the tools used in the workflow” on lne 369). We have also addressed the colloquial language highlighted by reviewer 2 (“huge”, “cleaning away”, “a little bit” and repetition of “for example”).

[L162 (caption 2.1.6)] now line 141 of the revised manuscript: numbering of this section has been altered to fit better with the numbering of the whole manuscript. This section heading has been updated to “Preparing metadata for analysis” in place of “Nice, neat metadata for analysis”.

A more thorough introduction to PCA, PERMANOVA and OPLS-DA has been provided with reference to other literature that covers this topic in more detail (lines 247-271 of the revised manuscript).

We hope the github repo that serves the website will continue to act as a forum for development and dissemination of guides and links to useful tutorials. We welcome further comment, suggestions and feedback from reviewer 2, and encourage them to contribute to the project via the github repo (https://github.com/LizzyParkerPannell/Untargeted_metabolomics_workflow).

To address the Minor review points:

  1. The figure sequence has been amended in the uploaded image files

  2. The capitalization in bullet points has been addressed to improve consistency throughout the manuscript.

Reviewer 3 Report

Please find my comments attached.

Author Response

We thank reviewer 3 for their considered appraisal of the manuscript and for their helpful suggestions. We would encourage reviewer 3 to take a look at the online guide at https://untargeted-metabolomics-workflow.netlify.app/ for some of the detailed instructions they allude to. 

In response to reviewer 3’s request for demo data to use to test the tools, section 2.9 (“Sharing metabolomics data”) of the revised manuscript highlights MetaboLights, which provides one route to accessible LCMS datasets that could be downloaded if a user would like to test the workflow. To support the workflow, we have made a MALDI training data set available through Dryad (https://doi.org/10.5061/dryad.dbrv15f5c ) and added links to this example data set and an existing LCMS data set for testing the workflow https://untargeted-metabolomics-workflow.netlify.app/00_overview/06_demo-data/ or training purposes.

We appreciate reviewer 3’s suggestion to make a video and include this within the online guide. This is something we hope to develop and include, though it requires some considerable planning to ensure its quality and accessibility for new users. We have, instead, focussed our time on revising the manuscript and existing content of the website.

  1. Section 2.1 has been renumbered in line with the rest of the manuscript. The section previously “2.1.1. Assumptions” has been removed for clarity.

  2. (Figure 2): caption has been altered to read “nitrogen” instead of “N” as advised.

  3. (Figure 2): caption has been altered to read “cell walls lysed and the sample homogenised” in response to reviewer 3’s comments.

  4. (Figure 2): The cartoon has been updated to better reflect an LC-MS set up (as opposed to MS only) as suggested by reviewer 3.

In response to the reviewer’s desire for further guidance on metabolite annotation, we would like to direct them to the associated website https://untargeted-metabolomics-workflow.netlify.app/07_putative-metabolite-id/01_what-are-my-metabolites/, where we have included instructions for using METLIN (as well as PubChem, MassBank and others) to undertake annotation and KEGG tools to sense check annotations.

We would welcome further feedback and contributions from reviewer 3 via the github repo (https://github.com/LizzyParkerPannell/Untargeted_metabolomics_workflow) once they have had a chance to test the workflow and guides with the example datasets.

Reviewer 4 Report

In the current manuscript, Parker et al. provide a detailed Mass spectrometry-based metabolomics workflow with experimental design, sample preparation, data acquisition, and analysis with statistical methods. All the information provided here is pretty much standard metabolomics practice in all metabolomic research labs except some specialized drug discovery workflows in pharmaceutical industries. There is no new information added to the workflow in the manuscript. However, this tutorial could be helpful for new researchers, non-experts, and students to get a sense of metabolomics workflow after careful rewriting.

I have listed a few points that may help to improve the manuscript.

1.    First of all, the authors need to structure the manuscript well to systematically write each step and provide the key points to consider while doing experiments or data analysis.

2.    The results section is actually the methods section, and the stages of workflow could be added as a table of content before the introduction.

3.    This workflow is fully based on vendor-specific mass spectrometer steps and needs to be made more universal considering the different operating conditions, mechanisms, raw data file formats, etc.  

4.    Section 2.1.3. Assumptions do not add any new information and may be converted to the mass spectrometer-specific requirements.

5.    Metabolite extraction and data acquisition section need to separate as sample preparation and metabolite extraction from different types of samples and data acquisition strategies. These sections are extremely important and need to be elaborated as it decides the outcome.

Finally, the data analysis and statistical analysis workflows need to be in great detail by explaining each step.

Author Response

We thank reviewer 4 for their careful consideration of the manuscript. We think that the workflow will be a helpful resource for new researchers as an entry point into metabolomics, as often workflows are for specific sample types or particular data processing streams. This workflow has been tested by several different research groups using multiple sample types and by users with varying levels of expertise in metabolomics. To make this clear within the manuscript, we have rewritten the introduction, in particular adding the following: “This resource is aimed at non-experts, and early career researchers who may not have extensive coding or analytical knowledge.” (lines 60-61 of the revised manuscript).

  1. The manuscript has been re-structured, with more clarity in the introduction and headings. 

  2. We have provided the stages of the workflow in a flow chart (figure 1) and a list in section 2, which we believe are clear and easy to follow and would like to suggest that this is sufficient to convey the information effectively. We feel that a table of contents would be superfluous.

  3. The manuscript has been edited to make clear that the workflow is based upon a universal format and is not specific to any particular vendor. The statement  “Proteowizard is capable of converting many other proprietary file formats and guidance is available through their extensive documentation at https://proteowizard.sourceforge.io/doc_users.html accessed 20th February 2023.” has been included in lines 182-184 of the revised manuscript. Earlier emphasis on the use of Waters/ MassLynx specific steps have been removed and mentioned only as examples so as not to detract from the interoperability of this workflow.

  4. The “Assumptions” section has been removed. Based on reviewers comments we felt it detracted from the flow of the manuscript and made the workflow seem specific to Waters data (when the workflow is in fact compatible with many vendor formats due to the conversion of data via Proteowizard to .mzML).

  5. In the online guide there is further detail and signposting on decision-making around which analytical approaches can be used (https://untargeted-metabolomics-workflow.netlify.app/02_data-acquisition/01_data-acquisition/). We briefly cover some of the reasons to choose different analytical techniques and link to further reviews and comparisons. The majority of researchers who come to metabolomics for the first time will be limited by the resources (equipment and expertise, but also funding) available in the lab or facility they have access to. In reality, there is rarely a free choice of which analytical approach to use. While most “wet lab” mass spectrometry facilities are maintained by dedicated technicians who can aid decision-making on analytical approaches, our experience is that ongoing bioinformatic support for external users of such facilities is often lacking or not included in run costs. We feel that the purpose of this manuscript is to guide the reader to explore open-source tools (via the online guide) in order to be able to deal with their data remotely, for free, and without access to technicians or bioinformaticians. Therefore we have avoided putting a lot of emphasis on the “wet lab” stages of the process in order to focus the manuscript on data processing and analysis. 

In response to reviewer 4’s comments on data analysis and statistical analysis, we have rewritten the explanations of the statistical approach covered in the workflow to include more detail on the use of PCA, PERMANOVA and OPLS-DA, as well as citing literature that elaborates these further (lines 247-271 of the revised manuscript). Furthermore, the online guide includes both R code and step-by-step instructions for undertaking both unsupervised and supervised analyses of datasets prepared using the workflow, with explanations of why each step is important. 

Round 2

Reviewer 1 Report

Unfortunately, the aim of the manuscript and the developed online resource is remained unclear. If this is an universal tool for analysis of untargeted metabolomics data, as the title suggests, then what does it have to do with the numerous mentions of plant metabolomics and section Metabolite exctraction? In such case the authors could simply offer their data as demo. If this is an example of success workflow for plant metabolomics investigation, then the manuscript title does not correspond to and it is unclear what the result is.

Author Response

We thank reviewer 1 for taking the time to read the revised manuscript and appreciate their feedback. We have addressed their concerns over the aim, objective and scope of the workflow with the following alterations to the manuscript:   L51-54 - we have added details of the systems for which this workflow was developed, and those it has been applied to "... primarily in the fields of environmental metabolomics and the study of complex plant stress responses. However, the tutorial and workflow have been applied in a range of experimental systems including E. coli, potato, barley, organic fertilisers, field soil samples, human cervical mucus, and Chlorella." L67 - we have added clarification on the output of the workflow "...to give a list of potential compounds that are differentially expressed between groups of samples which can inform the hypothesis for downstream targeted analyses." L89-91 - we have highlighted the availability of other datasets that the workflow can be applied to "An index of openly-available datasets is provided at https://untargeted-metabolomics-workflow.netlify.app/00_overview/06_demo-data/ (Accessed on 9 March 2023). These example datasets can be used to demonstrate the workflow presented here."   We are aware that reviewer 1 feels the section on "Metabolite extraction and data acquisition" is superfluous to the aims of the manuscript. It is our opinion that this section is important for readers who are new to the field of metabolomics in understanding the context of the workflow. In the previous round of revisions we included more detail on these steps in the online guide (for example at https://untargeted-metabolomics-workflow.netlify.app/02_data-acquisition/01_data-acquisition/) with the online guide being regularly signposted throughout the manuscript. As such, we have not removed the section "Metabolite extraction and data acquisition", but have swapped its order in the manuscript so that it appears prior to "2.4. Preparing metadata for analysis" to improve the flow of the text.

Reviewer 2 Report

The authors have responded to the issues raised previously to improve the manuscript. Overall, the manuscript is fit for publishing with the following minor improvements: 1. the manuscript objective formulated as the last topic in the introduction should be a single, not too long, last paragraph in the introduction. 2. The references were apparently extended manually, please double-check their correctness.

Author Response

We thank reviewer 2 for taking the time to read the manuscript again.

1) The manuscript objective has been reformulated as a single sentence at the end of the introduction, as advised by reviewer 2:

"This collaborative and open-source workflow guide for untargeted metabolomics addresses the need for data-handling tutorials [1] with the key aims of widespread use and continuous improvement, ultimately encouraging integration with multi-omic workflows."

2) The references have been double checked and are in order.

Reviewer 4 Report

The revised version looks much improved. The authors addressed all the issues and provided enough explanation in the manuscript.

Minor point:  Need to improve the quality of Figures 1 and 3.

Author Response

We thank reviewer 4 for taking the time to read the revised manuscript and appreciate their feedback. We have addressed their minor point about the quality of figures 1 & 3 by replacing those figures with higher resolution versions, both in the manuscript and the online upload of figures.

Round 3

Reviewer 1 Report

The authors have satisfactorily addressed most of my concerns.

Author Response

We thank reviewer 1 for taking the time to re-read the manuscript.